# Dynamic wrinkling pattern exhibiting tunable fluorescence for anticounterfeiting applications

Tianjiao Ma [1], Tiantian Li[1], Liangwei Zhou[1], Xiaodong Ma[1], Jie Yin[1] & Xuesong Jiang [1✉]

A dynamic surface pattern with a topography and fluorescence in response to environmental stimulus can enable information recording, hiding, and reading. Such patterns are therefore widely used in information security and anticounterfeiting. Here, we demonstrate a dynamic dual pattern using a supramolecular network comprising a copolymer containing pyridine (P4VP-nBA-S) and hydroxyl distyrylpyridine (DSP-OH) as the skin layer for bilayer wrinkling systems, in which both the wrinkle morphology and fluorescence color can be simultaneously regulated by visible light-triggered isomerization of DSP-OH, or acids. Acid-induced protonation of pyridines can dynamically regulate the cross-linking of the skin layer through hydrogen bonding, and the fluorescence of DSP-OH. On selective irradiation with 450 nm visible light or acid treatment, the resulting hierarchical patterned surface becomes smooth and wrinkled reversibly, and simultaneously its fluorescence changes dynamically from blue to orange-red. The smart surfaces with dynamic hierarchical wrinkles and fluorescence can find potential application in anticounterfeiting.

[1] School of Chemistry & Chemical Engineering, Frontiers Science Center for Transformative Molecules, State Key Laboratory for Metal Matrix Composite Materials, Shanghai Jiao Tong University, 200240 Shanghai, P.R. China. ✉email: ponygle@sjtu.edu.cn

Counterfeiting is a worldwide issue that has disastrous implications on the economy, human health, and national security. For example, part of all consumer goods, medicines and medical products, and even electronic components in military systems are counterfeits, which has a tremendous impact on our daily lives. Thus, anticounterfeiting technologies have found widespread applications in banknotes, diplomas, certificates, jewelry, medicines, and electronics[1–7]. Various graphically encoded taggants based on stimuli-responsive molecules have been developed to impede duplication because of their high coding capacity[8–19]. However, these are still replicable by counterfeiters due to their low complexity, predictable production process, and deterministic decoding mechanism. More complex tags with higher security often incur a high cost, which limits their application in consumer products. There is an increasing need for low-cost anticounterfeiting methods that cannot be replicated. In addition, the ideal encoded taggant must be stable, easily decodable, and suitable for mass production.

Among the various strategies available for anticounterfeiting, fluorescent patterns have been widely used in many fields and play an important role in anticounterfeiting technologies because of the readily detectable chemical characteristics involved[17–22]. The information security can be increased dramatically by introducing dynamic fluorescence that can be tuned by external stimuli to the surface pattern. A multifunctional fluorescent pattern in response to orthogonal stimuli was achieved by relying on light-triggered anthracene–endoperoxide and vapor-triggered monomer–polymer transitions, which lead to higher security reliability[19]. However, anticounterfeiting tags based on fluorescence patterns still face the risk of being cloned after the fluorescence compound is disclosed. In addition to the fluorescent pattern, surface wrinkles similar to those widely found on the skin of certain animals that are as unique as fingerprints, can be used as biomimetic fingerprints toward anticounterfeiting because of the similar minutiae to fingerprint, such as ridge ending and bifurcation[21–25]. In addition, wrinkles could also serve as graphical tags since the graphical images formed by wrinkles could be identified by naked eyes due to the light scattering caused by wrinkling pattern. Owing to the randomness, 3D topography, and nondeterministic process and unpredictability of the formation, wrinkling patterns caused by a surface mechanical instability[26–32] can realize a higher level of security in anticounterfeiting. Combining both responsive fluorescent behavior and the dynamic wrinkling pattern[33–48] into the same anticounterfeiting tag will undoubtedly enhance the information capacity and security. Owing to the complexity of the involved chemistry and material, however, it is still very challenging to fabricate surface patterns offering dynamic fluorescence and topography.

Herein, we demonstrate a feasible approach for generating a dynamic dual-function pattern exhibiting a wrinkled topography and fluorescence as fast response to multiple stimuli, such as visible light and acid gas based on a bilayer system, in which the skin layer is made up of a supramolecular polymer network and poly(dimethylsiloxane) (PDMS) serves as the soft substrate (Fig. 1). The supramolecular crosslinked network film composed of a copolymer-containing pyridine (P4VP-nBA-S) and hydroxyl distyrylpyridine (DSP-OH) exhibiting bright fluorescence was rigid enough to form wrinkles via hydrogen bonding between pyridine and hydroxyl groups. As the supramolecular crosslinking of hydrogen bonding in the top layer was sensitive to the photoisomerization of DSP[49–51] and acid gas[50–52], the wrinkled topography as well as the fluorescence could be simultaneously regulated by light and pH. This reversible dynamic pattern displaying wrinkles and fluorescence in response to visible light and acid gas can find potential application in anticounterfeiting and

information storage due to the advantages of multiple responses, region selectivity, and noncontact characteristics.

## Results

**Strategy of dynamic wrinkling pattern with tunable fluorescence.** The entire strategy for the fabrication of the dynamic wrinkled pattern with tunable fluorescence based on multi-responsive supramolecular network is illustrated in Fig. 1. The key point in the strategy is that the internal stress in the bilayer system or the modulus of the supramolecular network and the intensity or the wavelength of the fluorescence can be controlled simultaneously by reversible photoisomerization or protonation. The pyridine-containing copolymer (P4VP-nBA-S, $M_n = 16,300$, $M_w/M_n = 1.92$, wherein the molar ratio of 4-vinylpyridine, n-butyl acylate, and styrene was ~1:2:2) was synthesized through free radical copolymerization, and the incorporation of styrene and n-butyl acrylate was to tune the mechanical properties. The DSP-OH with bright blue fluorescence was synthesized through a one-step method. A detailed description of the synthesis and characterization of materials is provided in Supplementary Figs. 1−3.

DSP-OH showed a great response to multiple stimuli such as visible light or acid. The kinetics of the photoisomerization of the DSP-OH was traced by ultraviolet–visible spectroscopy (UV–vis) (Fig. 2a) and $^1$H NMR spectra (Fig. 2b). Upon irradiation under 450 nm light with an intensity of 15 mW cm$^{-2}$, the UV–vis absorption peaks of DSP-OH shifted to a lower wavelength, which was due to the weakening of the conjugated and planar structure of the Z-isomer (Fig. 2a). The $^1$H NMR spectra also provided strong evidence that the DSP-OH underwent photoisomerization under 450 nm light (Fig. 2b). It was clearly observed that new signals (1′−6′) appeared after irradiation, indicating a configuration change from the E-isomer to Z-isomer of DSP-OH. As a result, the UV–vis absorption and fluorescence emission spectra showed great changes. The quantum yield of DSP-OH ($1.0 \times 10^{-5}$ mol L$^{-1}$ toluene solution) decreased from 53.6% to 5.5% after irradiation by 450 nm light. Consequently, unlike the bright blue fluorescence of the E-isomer, the Z-isomer of DSP-OH was almost colorless.

Due to the strong push-pull electronic structure of DSP-OH and pyridine groups, the UV–vis absorption and fluorescence emission band shifted to a longer wavelength after protonation (Fig. 2c, d). With increase in the concentration of CF$_3$COOH, the maximum fluorescence emission wavelength of $1.0 \times 10^{-6}$ mol L$^{-1}$ DSP-OH toluene solution changed from 445 to 570 nm. The color space coordinates defined by the Commission Internationale de L'Eclairage (CIE) calculated from the fluorescence spectra showed a color change from blue to orange-red (Fig. 2e), which agreed well with the photographs shown in Supplementary Fig. 4. It was notable that the quantum yield of DSP-OH ($1.0 \times 10^{-5}$ mol L$^{-1}$ toluene solution) did not change significantly from 53.6% to 46.4% with and without CF$_3$COOH, respectively, indicating that there was almost no decrease in the intensity of the fluorescence with the switching of the fluorescence color.

A toluene solution of a mixture of P4VP-nBA-S and DSP-OH was spin-coated on a PDMS substrate as a top skin layer with a typical thickness of 100 nm. The top layer was a supramolecular crosslinked network and rigid enough to cause considerable mismatch in the modulus and thermal expansion ratio with the elastic PDMS substrate. A thermal treatment at 110 °C was undertaken to introduce the compressive stress into the system. Wrinkles occurred and minimized the total energy of the system when the bilayer system cooled to room temperature (Fig. 1c). On irradiating with 450 nm light, the wrinkles and fluorescence both disappeared because of release of the stress and the transition of

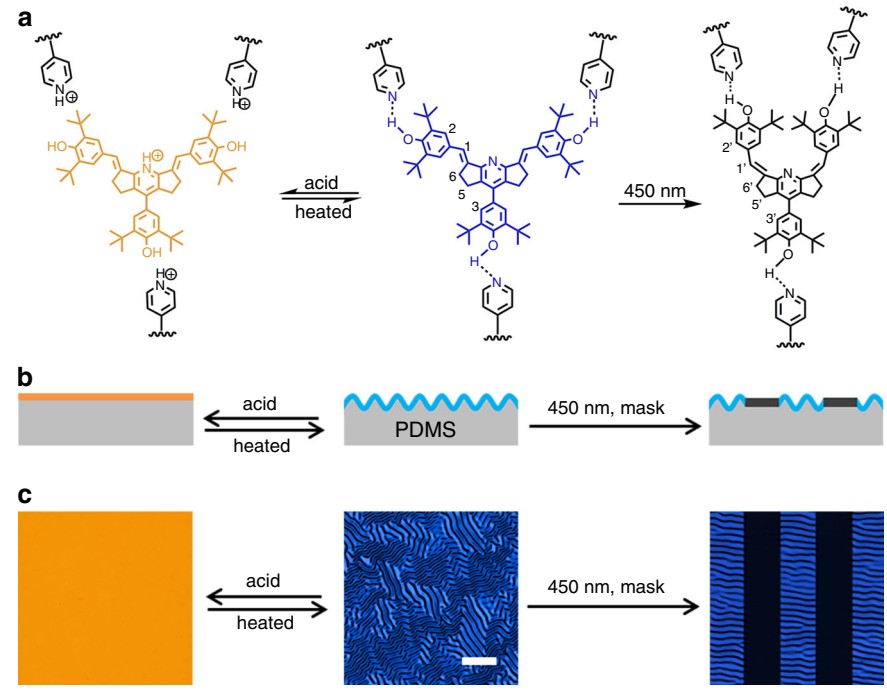

**Fig. 1 Strategy for the production of the dynamic wrinkled pattern with tunable fluorescence. a** Chemical structure of P4VP-nBA-S/DSP-OH served as the top layer and the protonation and photoisomerization reaction during the evolution process of winkled and fluorescent pattern. **b** Schematic illustration of the dual pattern with tunable wrinkle and fluorescence in response to visible light and acid based on (P4VP-nBA-S/DSP-OH)/PDMS bilayer system. **c** Corresponding laser scanning confocal microscope (LSCM) images of wrinkled surface with fluorescence color responsive to visible light and acid. Scale bar: 100 μm.

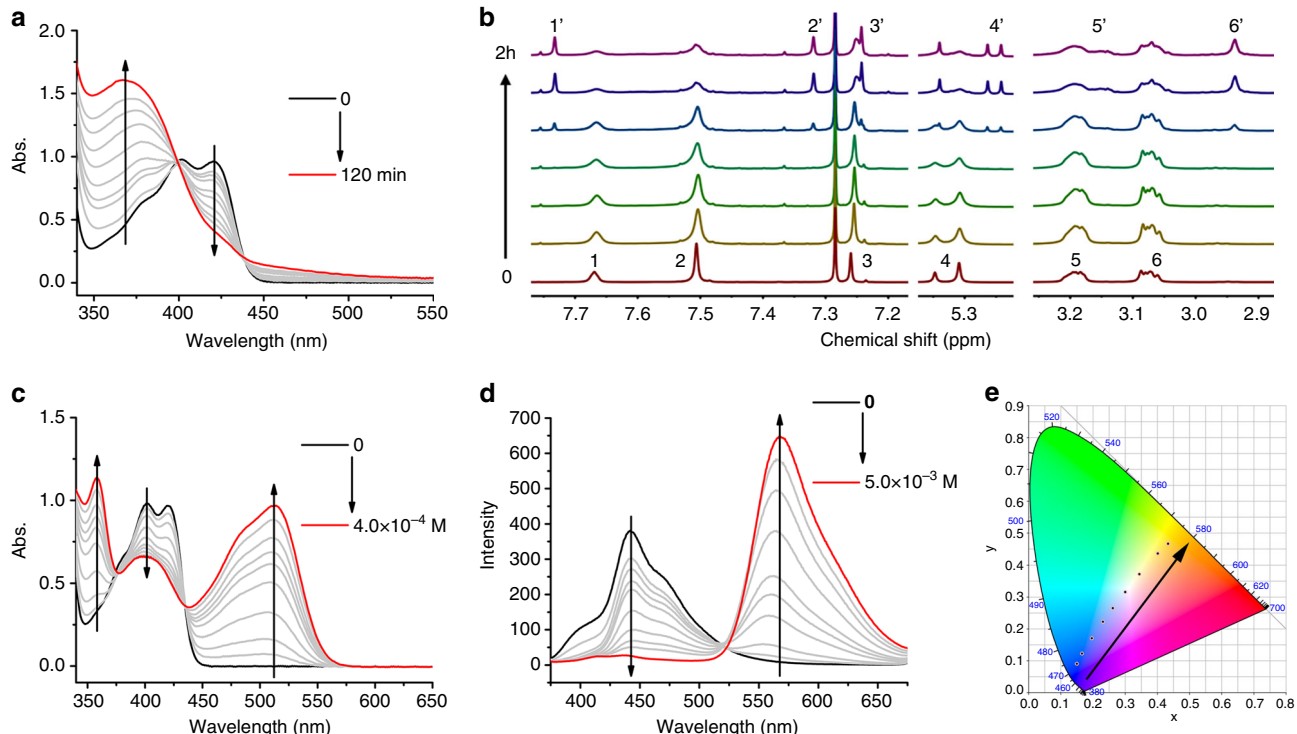

**Fig. 2 The photoisomerization and protonation reaction of DSP-OH. a** UV–vis spectra of $4.0 \times 10^{-5}$ mol $L^{-1}$ DSP-OH toluene solution for different irradiation times of 450 nm light. **b** $^{1}$H NMR spectra of DSP-OH for different irradiation times of 450 nm light in CDCl$_3$. **c** UV–vis spectra of $4.0 \times 10^{-5}$ mol $L^{-1}$ DSP-OH toluene solution with different concentration of CF$_3$COOH. **d** Fluorescence emission spectra of $1.0 \times 10^{-6}$ mol $L^{-1}$ DSP-OH toluene solution with different concentration of CF$_3$COOH. **e** The fluorescence color of $1.0 \times 10^{-6}$ mol $L^{-1}$ DSP-OH toluene solution with different concentration of CF$_3$COOH is illustrated in the CIE color space.

DSP-OH from the E-isomer form to the colorless Z-isomer, respectively (Fig. 1c). The wrinkles could also be erased by acid gas, such as hydrogen chloride (HCl) gas because the breakup of hydrogen bonding leads to decrosslinking and softening of the supramolecular network film and thus release of the internal stress. Protonation of DSP-OH also induced a color change from blue to orange-red due to the strong push-pull electronic effect in the chemical structure (Fig. 1c). The acid gas could be released through a thermal treatment so that the wrinkled and fluorescent patterns can be recovered.

We conducted a series of controlled experiments to investigate the factors for determining the formation and elimination of the wrinkled pattern. No obvious wrinkles were observed on the surface of individual P4VP-nBA-S or the mixture of P4VP-nBA-S and DSP-Bu (synthesized as shown in Supplementary Fig. 3)-coated PDMS after thermal treatment, indicating that the hydrogen bonding between pyridine and hydroxyl groups is essential for the generation of wrinkles (Supplementary Figs. 5a and 6). The occurrence of hydrogen bonding between pyridine and hydroxyl groups in the supramolecular network was confirmed by temperature-dependent FT-IR spectra (Supplementary Fig. 7). As the temperature increased, the peak assigned to the OH stretching vibration (3440 cm$^{-1}$) weakened and shifted to higher wavenumbers, which is the typical temperature-sensitive behavior of a hydrogen bond. According to the linear buckling theory, the size of the surface pattern is related to the modulus of the supramolecular film, which is determined by the density of hydrogen bonds in this system. The increase in the ratio of DSP-OH promoted the crosslinking density and mechanical properties of the top layer, consequently increasing the characteristic wavelength and amplitude, suggesting the significant role of hydrogen bonding in the formation of the wrinkled surface (Supplementary Fig. 5).

**Visible-light responsive wrinkle and fluorescence**. To gain detailed insight into the dependence of the wrinkled and fluorescent patterns upon the photoisomerization of DSP-OH, we monitored the morphological evolution of the surface pattern by atomic force microscopy (AFM, Fig. 3a), and traced the kinetics of the photoisomerization of the P4VP-nBA-S/DSP-OH film by UV–vis spectroscopy (Fig. 3c) and fluorescence spectra (Fig. 3d). The initial random and labyrinth wrinkles (Fig. 3a) exhibit that the distribution of minutiae such as ridge ending and bifurcation is nondeterministic[23–25], while the characteristic wavelength ($\lambda$) and amplitude ($A$) of wrinkles could be controlled by the mechanical properties of bilayer systems according to linear buckling theory[26–28]. The formation process is unpredictable, while the morphology of wrinkles locks in once they are formed. As shown in Fig. 3a, a sequence of 3D AFM images and corresponding fluorescence photographs under 450 nm light traced the evolution of wrinkles and fluorescence induced by the photoisomerization, suggesting the simultaneous control of the dual pattern. Efficient isomerization of the P4VP-nBA-S/DSP-OH film was demonstrated by UV–vis spectra in that the absorption peaks of DSP-OH shifted to a lower wavelength (Fig. 3c), which is similar to the reaction in solution. Owing to the photoisomerization under 450 nm light, the internal stress field of the wrinkled system underwent continuous disturbance, resulting in release of the internal stress and a rapid decrease in the characteristic amplitude ($A$) of the wrinkles. For instance, the amplitude decreased from 612 to 240 nm after irradiation for 10 min, and further irradiation led to complete elimination of the wrinkles. The detailed evolution process of the wavelength ($\lambda$) and amplitude ($A$) is shown in Fig. 3b. The amplitude ($A$) decreased with exposure time while the wavelength ($\lambda$) did not change significantly. In addition, the time-dependent fluorescence emission spectra of the P4VP-nBA-S/DSP-OH film showed that the fluorescence changed from bright blue to colorless (Fig. 3d), in accordance with the pictures in the inset of Fig. 3a, indicating the visible light regulation of the fluorescence along with the wrinkled topography.

The high spatial resolution and noncontact characteristics of the light-induced isomerization reaction provide possibilities for controlling spatial stress release, resulting in selective erasure of the wrinkled pattern. When a sample with the wrinkled surface was irradiated by 450 nm light through different photomasks, such as stripes, annuluses, or the letter "S", the wrinkles were selectively erased in the exposed regions, while the unexposed area remained wrinkled (Fig. 3e, Supplementary Fig. 8). The initially disordered wrinkles in the unexposed region became highly ordered and oriented perpendicular to the boundary of the exposed region, which might be ascribed to the boundary effect. We traced the evolution process of the wrinkles by laser scanning confocal microscopy (LSCM). As shown in Supplementary Fig. 9, the difference between the wrinkles in the exposed and unexposed regions became greater with increasing illumination time. Finally, the wrinkles in the exposed regions were fully erased, while the unexposed area remained wrinkled. Furthermore, owing to the great difference between the fluorescence of the two isomers, fluorescent micropatterns of stripes, annuluses, or the letter "S" were also obtained through various photomasks (Fig. 3e, Supplementary Fig. 8), for which the evolution process was observed by super resolution multiphoton confocal microscopy (STED). As shown in Supplementary Fig. 10, the fluorescence intensity difference for the exposed and unexposed regions increased rapidly when the sample was irradiated by 450 nm light. As a result, the exposed regions became dark while the fluorescence of the unexposed area remained blue. Therefore, the dual pattern with hierarchical wrinkles and fluorescence can be realized simultaneously by 450 nm irradiation with a photomask.

**pH-controlled wrinkling and fluorescence**. Hydrogen bonding between pyridine groups in P4VP-nBA-S and hydroxyl groups in DSP-OH endowed the dual pattern unique sensitivity to acid gas. In this study, HCl gas was employed to control the hydrogen bonding interaction. The protonation of pyridine groups weakened the hydrogen bond, resulting in decrosslinking of the supramolecular network, decrease in the modulus of the film, and release of the internal compressive stress of the bilayer system. Therefore, the wrinkled surface could turn to a wrinkle-free state upon HCl vapor treatment. To gain a detailed insight into the erasure process of wrinkles by HCl vapor, we traced the morphological change of the surface pattern in the atmosphere containing 56.4 ppm HCl vapor by AFM. As shown in Fig. 4a, the amplitude decreased from 614 to 128 nm after exposure for 60 s, and further treatment led to the complete erasure of the wrinkles. The detailed data on the wavelength ($\lambda$) and amplitude ($A$) upon the 56.4 ppm HCl vapor treatment time are shown in Fig. 4b. The amplitude ($A$) decreased rapidly with the treatment time while the wavelength ($\lambda$) did not change significantly. It should be noted that the concentration of HCl in the atmosphere played an important role in the erasure rate. For instance, it took only seconds to erase the wrinkles under 592 ppm HCl vapor (Supplementary Movie 1), while the erasure time increased up to half an hour when the pressure of the HCl vapor was 5.23 ppm. The smooth surface returned to the wrinkled state on thermal treatment. After evaporating the HCl by heating, the supramolecular network became crosslinked again so that the modulus of the film increased and wrinkles occurred (Fig. 4a). The wrinkled

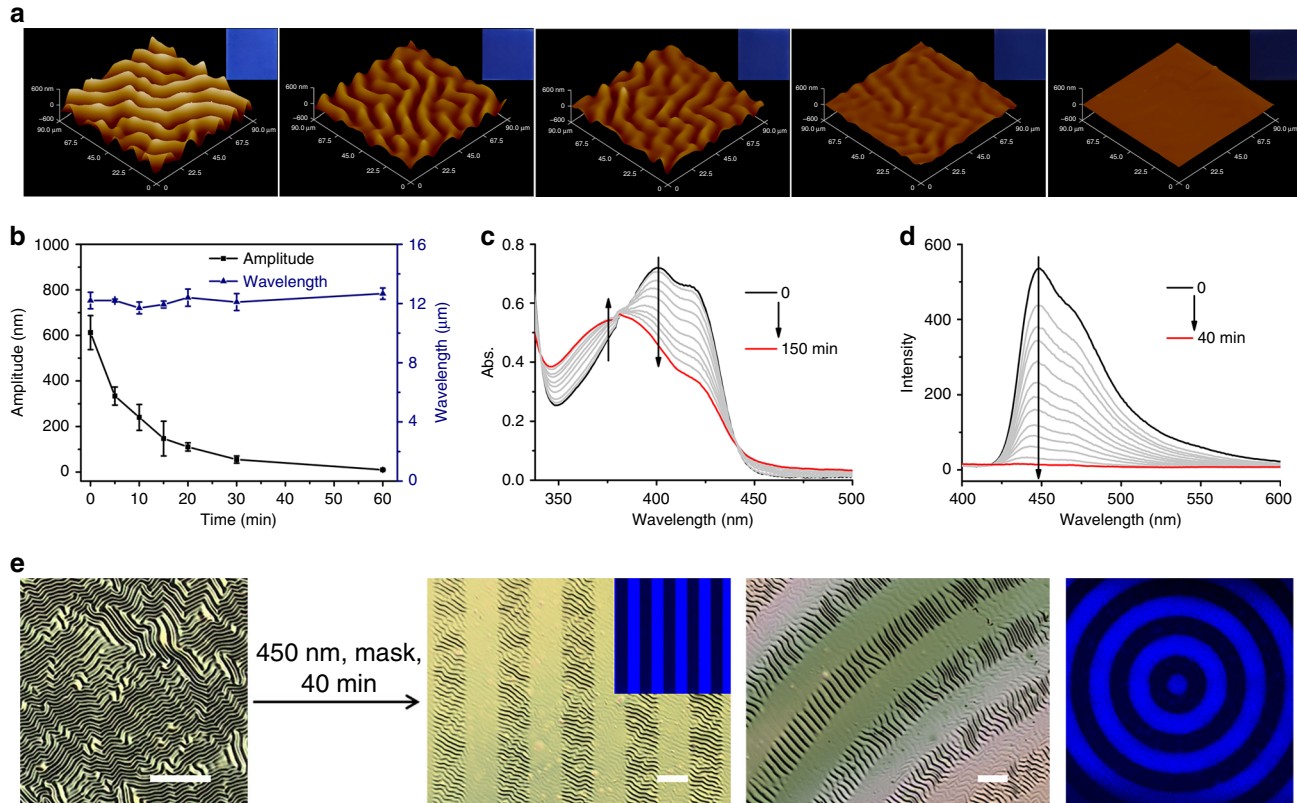

**Fig. 3 Evolution process of the dual-pattern under 450 nm light. a** 3D AFM images of wrinkles when the wrinkled samples were exposed to 450 nm light for 0, 5, 10, 20, and 60 min. Inset pictures are corresponding photographs of PDMS taken under UV light. **b** Amplitude (*A*, black square, left vertical axis) and wavelength (*λ*, blue triangle, right vertical axis) of the wrinkles as a function of 450 nm light irradiation time. Error bars represent the standard deviations of three independent data. Source data are provided as a Source Data file. **c** UV–vis spectra of P4VP-nBA-S/DSP-OH film for different irradiation times of 450 nm light. **d** Fluorescence emission spectra of P4VP-nBA-S/DSP-OH film for different irradiation times of 450 nm light. **e** LSCM images of strip/annulus wrinkled pattern and super resolution multiphoton confocal microscopy (STED) images of strip/annulus fluorescent pattern obtained by mask under 450 nm light for 40 min. Scale bar: 100 μm.

morphology was highly reversible for at least tens of cycles due to the dynamic nature of the hydrogen bonding (Supplementary Fig. 11a).

Furthermore, gradual protonation of the pyridine group in DSP-OH by HCl gas resulted in a noticeable but gradual change in the fluorescence from blue to orange. It was shown that the changes of UV–vis spectra and fluorescence emission spectra in the solid-state was similar to that in solution, indicating efficient protonation of the P4VP-nBA-S/DSP-OH film (Fig. 4c, d). The fluorescence emission spectra of the P4VP-nBA-S/DSP-OH film were recorded by fluorescence spectroscopy, showing a maximum emission wavelength change from 450 to 575 nm (Fig. 4d), and CIE color space coordinates were calculated from the fluorescence spectra (Fig. 4e). The color coordinates changed linearly from the blue region ($x = 0.1907$, $y = 0.2085$) to the orange ($x = 0.4292$, $y = 0.4094$) according to the CIE, indicating that the fluorescence color of sample can be well regulated by protonation of DSP-OH. Inset pictures in Fig. 4a visualize the evolution of fluorescence induced by 56.4 ppm HCl vapor, in agreement with the color change in CIE. Supplementary Movie 1 shows in situ observation of the fluorescence color under 592 ppm HCl vapor as well as the wrinkled topography. On heating to evaporate the HCl, the fluorescence color returned to the initial blue (Fig. 4a). The fluorescent pattern was highly reversible for at least tens of cycles due to the dynamic nature of the protonation (Supplementary Fig. 11b). Thus, simultaneous regulation of the wrinkled topography and fluorescence color was realized by acid treatment.

**Multilevel anti-counterfeiting technology and application**. The reversible dual pattern in response to light and acid can serve as a type of smart material that might find application in smart displays and message storage. The initial wrinkled surface was opaque under ambient conditions because of light scattering by the microscale wrinkled structures, and emitted blue fluorescence under UV light (Fig. 4f). As shown in Fig. 4f, a letter "Y" or "N" was written by a writing brush with a 5 wt% HCl solution. In the written area, the wrinkles flattened and were transparent, so that the letter "Y" or "N" could be identified by the naked eye. Under UV light, the written area displayed a pink "Y" or "N" while the other region emitted blue fluorescence. This dual pattern with wrinkles and fluorescence was reversible and could return to the wrinkled state with blue fluorescence on thermal treatment. Various dual-patterns can be realized through this simple approach (Supplementary Fig. 12), demonstrating the versatile and general method for application in smart displays.

Since it combines the wrinkled topography with fluorescence toward anticounterfeiting technologies, the multi-responsible dual-pattern will undoubtedly increase the information security. Further demonstration of application in message storage and anticounterfeiting is demonstrated in Fig. 5. The initial disordered wrinkles with blue fluorescence was obtained by a simple thermal treatment at 110 ℃ and subsequent cooling to room temperature of the (P4VP-nBA-S/DSP-OH)/PDMS system (Fig. 5a). Upon irradiation with 450 nm light through a QR code-shaped photomask, a wrinkled pattern of the QR code with blue fluorescence was realized, which carried the given information

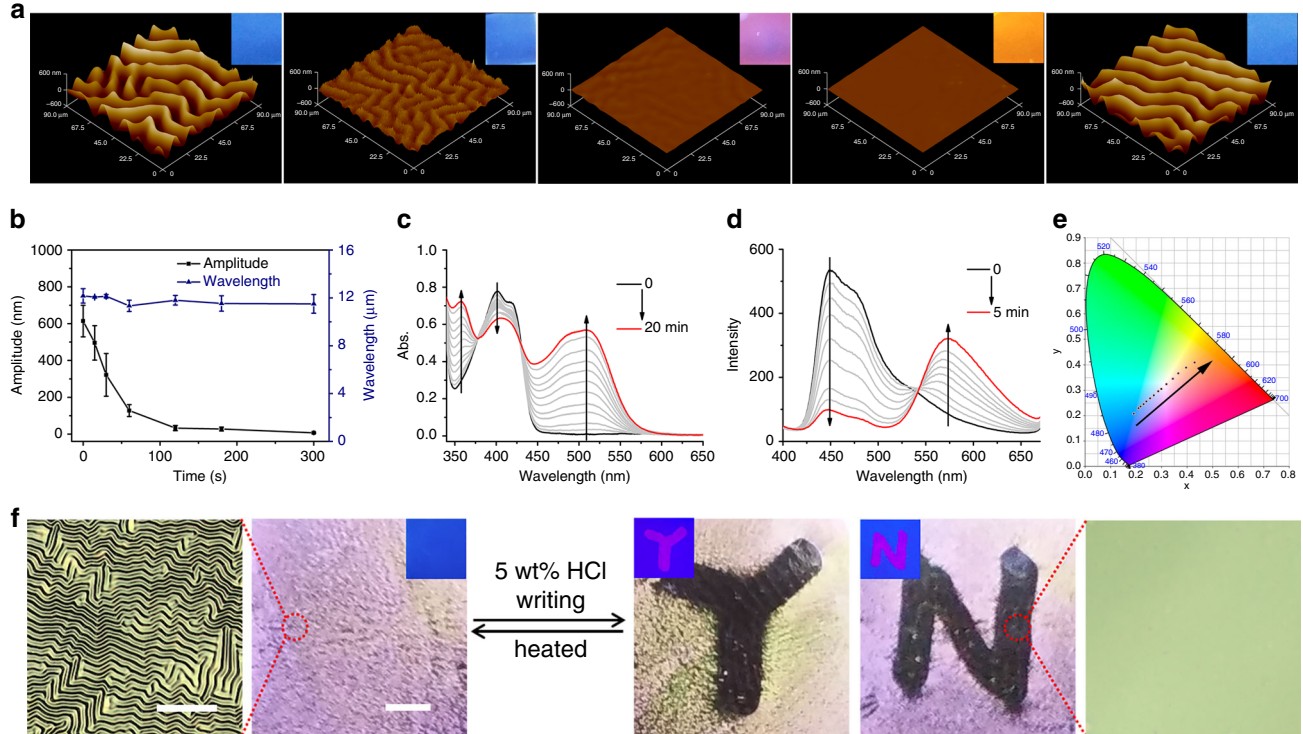

**Fig. 4 Evolution process of the dual-pattern on HCl vapor treatment. a** 3D AFM images of wrinkles when the wrinkled samples were exposed to 56.4 ppm HCl vapor for 0, 60, 180, and 300 s, and subsequently reheated to release the HCl. Inset pictures are corresponding photographs of PDMS taken under UV light. **b** Amplitude (*A*, black square, left vertical axis) and wavelength (*λ*, blue triangle, right vertical axis) of the wrinkles as a function of 56.4 ppm HCl vapor treatment time. Error bars represent the standard deviations of three independent data. Source data are provided as a Source Data file. **c** UV–vis spectra of P4VP-nBA-S/DSP-OH film for different times of 56.4 ppm HCl vapor treatment. **d** Fluorescence emission spectra of P4VP-nBA-S/DSP-OH film for different times of 56.4 ppm HCl vapor treatment. **e** The fluorescence color of P4VP-nBA-S/DSP-OH film at different HCl vapor treatment times illustrated in the CIE color space. **f** Photographs of letters "Y" and "N" on PDMS under UV light and natural light written by a writing brush with 5 wt% HCl. Scale bar: 2 mm. The corresponding LSCM images exhibiting characteristic wrinkled and flat surface. Scale bar: 100 μm.

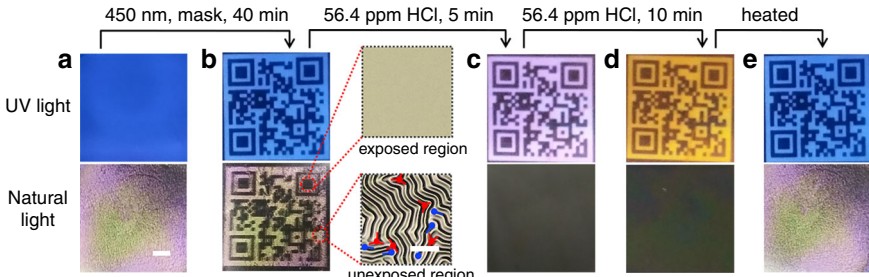

**Fig. 5 Photographs of QR code based on wrinkled and fluorescent pattern for anticounterfeiting. a** Initial labyrinth wrinkles and blue fluorescence. Scale bar is 2 mm. **b** QR code shaped wrinkled and blue fluorescent pattern obtained by a photomask of QR code under 450 nm light for 40 min; The corresponding LSCM images (right) exhibiting characteristic wrinkled and flat surface with or without minutiae, such as ridge ending (blue color) and bifurcation (red color). Scale bar is 50 μm. **c** Flat and QR code shaped purple fluorescent pattern that the sample **b** undergoing 56.4 ppm HCl for 5 min. **d** Flat and QR code shaped orange fluorescent pattern that the sample **c** undergoing 56.4 ppm HCl for 10 min. **e** Labyrinth wrinkled and QR code shaped blue fluorescent graphic images that the sample **d** undergoing thermal treatment.

(Fig. 5b). In addition, the individual minutia of wrinkle pattern such as ridge ending and bifurcation observed by laser scanning confocal microscope exhibit unique identifier, like a fingerprint, which confirms the uniqueness of the QR code pattern. The QR code graphical pattern and the fingerprint-like structures work together as a multilevel anticounterfeiting tag. When the sample was exposed to 56.4 ppm HCl vapor for 5 min, the wrinkles flattened and the fluorescence of the QR code changed to purple (Fig. 5c), showing the discrepant information that will increase the information capacity. Further exposure led to orange fluorescence of the QR code (Fig. 5d). After heating to evaporate

the HCl, the wrinkles occurred again and the fluorescence of the QR code returned to blue (Fig. 5e). Moreover, a human face-shaped wrinkled and fluorescent pattern was obtained by irradiating the initial sample with 450 nm light through a photomask of a human face (Supplementary Fig. 13). The pink nose and mouth were drawn using a writing brush with a 5 wt% HCl solution. After thermal treatment, labyrinthine wrinkles occurred and the nose and mouth with pink fluorescence disappeared because of evaporation of the adsorbed HCl. Thus, the multi-responsible dual pattern is substantially more difficult to be cloned compared to the single dynamic patterns responsive

to multiple stimuli or dual patterns operated in a single-mode fashion. Furthermore, multi-responsible strip and maple leaf patterns with wrinkles and fluorescence were obtained by the same method (Supplementary Figs. 14 and 15), proving the reliability and versatility of the method and the increased safety of the information.

## Discussion

In summary, we demonstrated a facile and robust strategy for fabricating a reversible and multi-responsible dual pattern exhibiting a simultaneously dynamic wrinkled topography and fluorescence based on a supramolecular network containing P4VP-nBA-S and DSP-OH. Both the fluorescence and wrinkled topography were orthogonally modulated via the stimuli of visible light and acid gas. The elimination of wrinkles was the consequence of stress release via photoisomerization of the DSP-OH or breakup of the dynamic crosslinked network by acid gas, while the reversible fluorescence change was caused by photo-isomerization or protonation of the DSP-OH. Owing to the spatial and simultaneous control of wrinkles and fluorescence by stimuli such as visible light and acid gas, the smart surface could be potentially employed in smart displays, information storage, and anticounterfeiting. Hence, the multi-responsible dual pattern based on a supramolecular network provides an effective strategy toward high-performance anticounterfeiting materials with high security reliability, dynamic characteristics, and ease of preparation.

## Methods

**Preparation of PDMS substrate**. The PDMS elastic sheet was prepared by mixing PDMS prepolymer (Sylgard 184, Dow Corning) in a 10:1 base/curing agent ratio, followed by drop-coating in a Petri dish, degassing in a vacuum oven, and curing at 70 °C for 4 h (thickness ~400 μm). Then the sample was cut into 1 cm × 1 cm and 2 cm × 2 cm squares.

**Preparation and erasure of wrinkle pattern**. A toluene solution of P4VP-nBA-S (3 wt%) and DSP-OH (from 0.38 to 1.5 wt%) was spin-coated onto a PDMS sheet to prepare the skin layer. The bilayer samples with fluorescent pattern were heated at 110 °C. When cooling to room temperature, wrinkled pattern occurred. For erasure of the wrinkled and fluorescent pattern, the samples underwent treatment with 450 nm light or HCl gas.

Details of material design, analysis, and instruments can be found in the Supplementary Information.

## Data availability

Data supporting the findings of this study are available within the paper and its Supplementary Information files. The source data underlying Figs. 3b, 4b and Supplementary Figs. 10b, 11a are provided as a Source Data file. All other relevant data that support the findings of this study are available from the corresponding author upon request.

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

## Acknowledgements
The authors thank the National Nature Science Foundation of China (51773114, 21704062) and the Shanghai Municipal Government (17JC1400700) for their financial support.

## Author contributions
X.J., X.M., and J.Y. conceived the research and analyzed the results and data; T.M. carried out the material synthesis and characterization; T.L. and L.Z. took part in some work of material synthesis. All authors contributed to the manuscript.

## Competing interests
The authors declare no competing interests.
