## [Peer Review File · Nature Communications]

Reviewers' comments:

Reviewer #1 (Remarks to the Author):

In this manuscript, the authors demonstrated a reversible and multi-responsive dual pattern exhibiting a simultaneously dynamic wrinkled topography and fluorescence based on a supramolecular network containing pyridine (P4VP-nBA-S) and hydroxyl distyrylpyridine (DSP-OH). Both the wrinkle morphology and fluorescence color can be simultaneously regulated by either visible light-triggered isomerization of DSP-OH, or acids such as HCl. Owing to the control of wrinkles and fluorescence on the skin layer, the smart surface could be potentially employed in smart displays, information storage, and anticounterfeiting. This manuscript can be accepted for publication after considering the following minor comments:

1. It should be "which is determined by the density of hydrogen bonds in this system" in line 165.
2. The word "of" should be deleted in line 178 because "visualize" is a transitive verb.
3. The annotation of Figure S6 in the supporting information should be "Inset picture is corresponding photographs..." because there is only one inset picture in Figure S6.
4. The units of the vapor treatment time in Figure 3(c) and 3(d) should be the same.
5. The following recently published important reference should be cited here "Various graphically encoded taggants based on stimuli-responsive molecules have been developed to impede duplication because of their high coding capacity": Mater. Chem. Front., 2017,1, 167-171.

Reviewer #2 (Remarks to the Author):

Author demonstrated a dynamic wrinkled and fluorescent pattern using a supramolecular network comprising a copolymer containing pyridine (P4VP-nBA-S) and hydroxyl distyrylpyridine (DSP-OH) as the skin layer for bilayer wrinkling systems. The dynamic surface pattern and capacity for exhibiting fluorescence in response to environmental stimulus can enable information recording, hiding, and reading.

In detail, author addressed that acid-induced protonation of pyridines in DSP-OH and P4VP-nBA-S can dynamically regulate the cross-linking of the skin layer through hydrogen bonding between hydroxyl and pyridine groups and the fluorescence color of DSP-OH. On selective irradiation with 450 nm visible light or acid treatment, the resulting hierarchical patterned surface became smooth and wrinkled reversibly, and simultaneously its fluorescence changed dynamically from blue to orange-red. Combining both responsive fluorescent behavior and the dynamic wrinkling pattern into the same anticounterfeiting tag will enhance the information capacity and security. Therefore, such pattern can be used in information security and anti-counterfeiting technologies.

Author concluded that the multi-responsive dual pattern based on a supramolecular network provides an effective strategy toward high-performance anticounterfeiting materials with high security reliability, dynamic characteristics, and ease of preparation

Reviewer think that the authors' wrinkled surface is produced by dry heating consisting of a soft polymeric layer coated with a stiff layer. The soft polymeric layer shrinks as the PDMS dry, whereas the stiff layer (P4VP-nBA-S/DSP-OH) does not. This generates excess surface area for the stiff layer, which wrinkles spontaneously to accommodate the shrinking soft polymeric layer. Many previous work studying the wrinkling of thin coatings on soft substrates reported labyrinth patterns similar to those described in the current paper. The fluorescence color changes is not suitable for use in anticounterfeiting tools.

The author also shows that a responsive fluorescent behavior and the dynamic wrinkling pattern as multilevel anti-counterfeiting technology.

However, it does not have any information inside the wrinkle pattern as an identifier. Although the labyrinth of the wrinkle pattern is deterministic, the meandering structure of each labyrinth is random and contains minutiae that can function as identifiers. The author needs to consider these factors, not QR-code, when designing new anti-counterfeiting technology.

Therefore, authors are required to consider the wrinkle pattern as an identifier in designing new anti-counterfeiting technology and also show that the wrinkle pattern can be deciphered. Overall, this paper is not considered a new anticounterfeit technique. I do not recommend its publication

Reviewer #3 (Remarks to the Author):

In this manuscript, the authors describe some interesting work on dynamic modulation of isotropic/anisotropic and spatial wrinkle topography as well as simultaneous fluorescence pattern based on a supramolecular network and photoisomerization chemistry.

The stimulus-responsive composite material is new compared to previous works in this group and the patterning method, the authors developed, is very simple but precise, applicable to various materials. The responsiveness of the involved photo-responsive supramolecular network and the tunability of the resulted reversible pattern are very nice. And these results are very interesting whatever from the perspective of the material chemistry, means to demonstrate diverse topological microstructures (such as macro-scale calligraphy pattern, QR pattern) or various potentials for smart displays, information storage, and anticounterfeiting of dynamic pattern indicated by authors.

In the end, I like the concise words and nice figures in the paper and I think it should be published. Hence, I recommend this manuscript for publications in Nature Communications.

Some nitpicks:

- 1) I am curious why the CIE color space coordinates changed linearly demonstrated in Figure 3e. The authors mentioned it while seem to have no discussion on this interesting result.
- 2) some scale bars of Figure 3f and Figure 4 look like absent, and I think the writing calligraphy and QR patterns should be macro-scale ones.
- 3) in Figure 2e, the exposure time of 450 visible light under the photomask is unclear both in main text and annotation in figure.
- 4) there is a typo error "wrinting" in Figure 3f.

Reviewer #1: In this manuscript, the authors demonstrated a reversible and multi-responsive dual pattern exhibiting a simultaneously dynamic wrinkled topography and fluorescence based on a supramolecular network containing pyridine (P4VP-nBA-S) and hydroxyl distyrylpyridine (DSP-OH). Both the wrinkle morphology and fluorescence color can be simultaneously regulated by either visible light-triggered isomerization of DSP-OH, or acids such as HCl. Owing to the control of wrinkles and fluorescence on the skin layer, the smart surface could be potentially employed in smart displays, information storage, and anticounterfeiting.

Response: We thank you for the very positive comments. The manuscript has been carefully revised according to the suggestions.

Comment 1: *It should be “which is determined by the density of hydrogen bonds in this system” in line 165.*

Response to 1: Thank you for your good advice. We have modified the manuscript to correct the unsuitable sentences.

Comment 2: *The word “of” should be deleted in line 178 because “visualize” is a transitive verb.*

Response to 2: Thanks for your careful check. Some revisions have been made in text (page 8, line 5) accordingly .

Comment 3: *The annotation of Figure S6 in the supporting information should be “Inset picture is corresponding photographs...” because there is only one inset picture in Figure S6.*

Response to 3: Thanks for your careful check. We have modified the manuscript and the supporting information to correct the unsuitable sentences and terminologies.

Comment 4: *The units of the vapor treatment time in Figure 3(c) and 3(d) should be the same.*

Response to 4: Thanks for your good suggestions. We have revised these figures with

better resolution in the revised manuscript.

Comment 5: *The following recently published important reference should be cited here “Various graphically encoded taggants based on stimuli-responsive molecules have been developed to impede duplication because of their high coding capacity”: Mater. Chem. Front., 2017,1, 167-171.*

Response to 5: Thank you for kindly reminding us this paper. We agree that there are numerous publications fluorescent patterns on polymeric surface that was not cited. However, this is not a review article. Dynamic patterns were discussed in some of the previous work, while there was not a detailed understanding. We already cited numerous studies (ref. 8-19) and, as a consequence of the referee’s criticism, we have included references to works of Wu and Huang (*Mater. Chem. Front.* **2017**, 1, 167-171) in the revised manuscript.

Reviewer #2: Author demonstrated a dynamic wrinkled and fluorescent pattern using a supramolecular network comprising a copolymer containing pyridine (P4VP-nBA-S) and hydroxyl distyrylpyridine (DSP-OH) as the skin layer for bilayer wrinkling systems. The dynamic surface pattern and capacity for exhibiting fluorescence in response to environmental stimulus can enable information recording, hiding, and reading.

In detail, author addressed that acid-induced protonation of pyridines in DSP-OH and P4VP-nBA-S can dynamically regulate the cross-linking of the skin layer through hydrogen bonding between hydroxyl and pyridine groups and the fluorescence color of DSP-OH. On selective irradiation with 450 nm visible light or acid treatment, the resulting hierarchical patterned surface became smooth and wrinkled reversibly, and simultaneously its fluorescence changed dynamically from blue to orange-red. Combining both responsive fluorescent behavior and the dynamic wrinkling pattern into the same anticounterfeiting tag will enhance the information capacity and security. Therefore, such pattern can be used in information security and anti-counterfeiting technologies.

Response: We thank you for the insightful comments. It inspires us very much and the manuscript has been carefully revised according to the suggestions.

Comment 1: *Author concluded that the multi-responsive dual pattern based on a supramolecular network provides an effective strategy toward high-performance anticounterfeiting materials with high security reliability, dynamic characteristics, and ease of preparation. Reviewer think that the authors' wrinkled surface is produced by dry heating consisting of a soft polymeric layer coated with a stiff layer. The soft polymeric layer shrinks as the PDMS dry, whereas the stiff layer (P4VP-nBA-S/DSP-OH) does not. This generates excess surface area for the stiff layer, which wrinkles spontaneously to accommodate the shrinking soft polymeric layer. Many previous work studying the wrinkling of thin coatings on soft substrates reported labyrinth patterns similar to those described in the current paper.*

Response to 1: Thank you for your insightful consideration. We agree that tunable and reversible wrinkle pattern is fascinating and it is no doubt that there are numerous

publications on wrinkle patterns on polymeric surface. And the dynamic wrinkle patterning was discussed in some of the previous works (ref. 33-48). However, this paper is markedly different from those reported works. Our purpose is to present a dynamic dual pattern exhibiting both a wrinkled topography and fluorescence in response to multiple stimuli such as visible light and acid, which was not reported and realized in the previous study.

Actually, the phenomenon of self-wrinkling is widespread in nature, such as dried apple and old man's face, and the theory of bilayer wrinkling system has been studied for decades. Many previous works are based on the bilayer system. However, the exact preparation methods are various, individual and fascinating, resulting in unique properties and widespread applications, such as tunable optical devices, flexible electronic devices, switchable wettability and smart display. Some wrinkled patterns in response to multi stimulus, or dual patterns operated in one mode are fabricated. But it is still a challenge for the multi-responsive dual pattern. We demonstrated a simple and feasible approach for generating a dual pattern in response to multiple stimuli, taking advantages of the supramolecular polymer network, photochemistry and bilayer wrinkling system. Just as commented by reviewer #3, the stimulus-responsive composite material is new compared to previous works. In addition, the strategy for multi-responsive dual pattern with wrinkled topography and fluorescence is novel. The patterning method is very simple but precise, applicable to various materials. The results are very interesting whatever from the perspective of the material chemistry, which means to demonstrate diverse topological microstructures, or various potentials for smart displays, information storage, and anticounterfeiting of dynamic pattern.

Also, the wrinkling mechanism in this work is not entirely sure as the referee comments. The internal stress comes from the mismatch of shrinkage between the PDMS substrate and the stiff skin layer when the system cool down, but not "as the PDMS dry". During the cooling process, the surface wrinkles were formed spontaneously. The minutiae of the labyrinth wrinkle such as ridge ending and bifurcation is nondeterministic, while the characteristics such as wavelength and amplitude are determined by the bilayer system. Previous study revealed that each

wrinkled pattern has different minutiae, linking the natural fingerprints with wrinkles (*Adv. Mater.* **2015**, 27, 2083-2089; *Sci. Adv.* **2017**, 3, e1700071; *Nature* **2015**, 520, 164-165). Furthermore, although the formation process is unpredictable, the wrinkling patterns lock in once they are formed. Thus, wrinkles can serve as artificial fingerprints, because it exhibit the similar encoded method and hidden information with fingerprints. On the other hand, the wavelength and the amplitude of wrinkle is determined by the mechanical properties of bilayer systems according to linear buckling theory (*Science* **2006**, 311, 208-212; *Nature* **1998**, 393, 146-149; *Acc. Chem. Res.* **2019**, 52, 1025-1035). The amplitude and the wavelength of wrinkle is given as Equation E1 and E2

$$A = h_f \sqrt{\frac{\varepsilon - \varepsilon_c}{\varepsilon_c}} \quad (\text{E1})$$

$$\lambda = 2\pi h_f \left(\frac{E_f}{3E_s} \right)^{2/3} \quad (\text{E2})$$

Consequently, the characteristics of wrinkle can be controlled by tuning the thickness of the film, the modulus of the film and the substrate, and the strain in the system. In addition, the dynamic chemistry endows wrinkled pattern dynamic characteristic, whose elimination and recovery of the wrinkles can be controlled on demand. In our present system, the obtained wrinkles could be eliminated due to the photoisomerization of DSP-OH induced release of the internal stress, which is a new chemistry in generation of dynamic wrinkle. So the wrinkled topography could be dynamic tuned precisely by visible light. Moreover, owing to the high spatial resolution of light, photoreaction provides a strategy for the region selective elimination of wrinkles, causing a difference of optical property between the wrinkled and flat area. Thus, wrinkles could also serve as graphical tags because the graphical images formed by wrinkles could be identified by naked eyes due to the light scattering of wrinkled surface. In summary, the wrinkled pattern is controlled and tuned dynamically as we demonstrated in the manuscript, with the individual minutiae.

Comment 2: *The fluorescence color changes is not suitable for use in anticounterfeiting tools.*

Response to 2: Generally, fluorescent image is widely used in anticounterfeiting field. Moreover, the reversible fluorescence color changes are supposed to a potential alternative for use in anticounterfeiting tools. Generally speaking, fluorescent anticounterfeiting technologies have been playing an important role in our daily life widely used in many fields, such as banknotes. To further reduce the risk of being cloned and enhance the information security, great efforts have been made to develop dynamic and reversible fluorescent patterns. There are numerous publications on fluorescent patterns for anticounterfeiting. For instance, Arppe and coworkers presented a series of “Physical unclonable functions generated through chemical methods” (*Nat. Rev. Chem.* **2017**, 1, 0031), including the fluorescent tags; Gao and coworkers presented a work about fluorescent anti-counterfeiting through “Cooperative supramolecular polymers with anthracene–endoperoxide photo-switching” (*Nat. Commun.* **2018**, 9, 3977); Qi and coworkers developed a “Solid-State Photoinduced Luminescence Switch” (*J. Am. Chem. Soc.* **2017**, 139, 16036-16039); Hou and coworkers generated “Tunable solid-state fluorescent materials for supramolecular encryption” (*Nat. Commun.* **2015**, 6, 6884). In addition, there are a variety of works that can not be list limited by the space (*Nat. Commun.* **2019**, 10, 2409; *Angew. Chem. Int. Ed.* **2019**, 131, 8865-8870; *Adv. Mater.* **2017**, 29, 1605271; *Angew. Chem. Int. Ed.* **2019**, 58, 17814-17819). Therefore, the reversible fluorescence color changes are suitable for use in anticounterfeiting tools, which can effectively increase the level of anticounterfeiting due to the hidden information in the dynamic reversible process.

Comment 3: *The author also shows that a responsive fluorescent behavior and the dynamic wrinkling pattern as multilevel anti-counterfeiting technology. However, it does not have any information inside the wrinkle pattern as an identifier. Although the labyrinth of the wrinkle pattern is deterministic, the meandering structure of each labyrinth is random and contains minutiae that can function as identifiers. The author needs to consider these factors, not QR-code, when designing new anti-counterfeiting*

technology. Therefore, authors are required to consider the wrinkle pattern as an identifier in designing new anti-counterfeiting technology and also show that the wrinkle pattern can be deciphered. Overall, this paper is not considered a new anticounterfeiting technique. I do not recommend its publication.

Response to 3: We thank the referee for the comment, though we beg to differ with the referee. In fact, the wrinkled pattern exhibits potential application in multilevel anticounterfeiting, combining the characteristics and advantages of traditional graphical tags and fingerprint-like structures (Figure R1). On one hand, the graphical images formed by wrinkles could be identified by naked eyes due to the light scattering of wrinkle pattern, which has the similar characteristic with graphical encoded tags, such as raised print, watermarks, fluorescent pattern and hologram (*Nat. Rev. Chem.* **2017**, 1, 0031), often found on banknotes, diplomas and certificates. As for these tags, the displayed images play the role of identifiers, and wrinkles are no exception. However, these tags still face the risk of being cloned even without the exact compound and method. To enhance the information security, great efforts have been made to develop dynamic patterns. On the other hand, wrinkles have the individual minutiae, because of the fingerprint-like structures, which is the advantage over other graphical anticounterfeiting tags (*Nat. Rev. Chem.* **2017**, 1, 0031). Previous study revealed that wrinkle can be used as man-made fingerprint due to the similar minutiae such as ridge ending and bifurcation, and identification (*Adv. Mater.* **2015**, 27, 2083-2089; *Nature* **2015**, 520, 164-165). Thus, minutiae of wrinkled pattern such as ridge ending and bifurcation can serve as the identifier, too, which is working with the visible graphical images that could be identified by naked eyes due to the light scattered by wrinkle referred before, and as a result, enhancing the information security. Figure R1 shows the similarity between analogous dynamic fingerprints and wrinkles as artificial fingerprints. Moreover, Figure R1b exhibits the multilevel characteristics of both visible graphical tags and fingerprint-like microstructures. Therefore, wrinkles show great potential for multilevel anticounterfeiting, because it combines the advantages of graphical tags and fingerprint-like structures, and dynamic fluorescence.

Figure R1. Schematic illustration of fingerprints (a) and wrinkles as dynamic artificial fingerprints with tunable fluorescence (b), exhibiting the similar encoded method and hidden information between them, and the potential application in multilevel anticounterfeiting.

Anti-counterfeiting is a hot topic on information protection since counterfeiting has brought serious negative impacts in almost every aspect in our daily life. Fluorescent patterns have been used in many fields for anticounterfeiting technologies because of the readily detectable chemical characteristics involved. Surface wrinkles exhibit potential application in anticounterfeiting due to its graphical pattern and biomimetic fingerprint-like structure. Combining the advantages of fluorescent and wrinkled pattern will undoubtedly enhance the information capacity and security, but it is still a challenge due to the complexity of the involved chemistry and material. Thus, in this article, we introduce a novel and feasible anticounterfeiting method through generating a multi-responsive dual pattern. Through this strategy, we have achieved a tag with multiple patterns and operated in multiple modes, consequently, editable and tunable encoded and decoded method, as shown in Fig. 5 and Supplementary Figs. 12-15. Compared to previous works, the stimulus-responsive composite material is new and the pattern exhibits more interesting properties. It is a progress in the dynamic surface due to its dual pattern and the multiple operating mode, as shown in Table R1. Moreover, although the pattern is more complex, the patterning method demonstrated in this work is very simple but precise, applicable to various materials, as referred to reviewer #3.

Table R1. Some of the present wrinkled or fluorescent patterns for anticounterfeiting tags

Tag	Operating mode	Reference
wrinkle	none	Adv. Mater. 2015 , 27, 2083-2089 Nature 2015 , 520, 164-165
fluorescence	dual	Nat. Commun. 2018 , 9, 3977
wrinkle and fluorescence	single	ACS Materials Lett. 2019 , 1, 77-82
wrinkle and fluorescence	dual (visible light and acid)	this work

The topic of our work is to present a versatile strategy for generating multi-responsive dual pattern, and display the potential value and application by some examples (Fig. 5, Supplementary Figs. 12-15). In fact, either the tunable fluorescence or the dynamic wrinkled pattern exhibits unique properties for a new alternative and method in counterfeiting. But we don't think we need to do much more work that is supposed to do by engineers, such as applying it to the pack or certificate. There is no doubt that some limitations exist and efforts are needed to improve the system. We will optimize the structure of the molecular and design more sensitive system in the follow works, to improve the application in counterfeiting. As a result of the referee's criticism, we have added some discussion and revised the manuscript carefully accordingly.

Reviewer #3: In this manuscript, the authors describe some interesting work on dynamic modulation of isotropic/anisotropic and spatial wrinkle topography as well as simultaneous fluorescence pattern based on a supramolecular network and photoisomerization chemistry.

The stimulus-responsive composite material is new compared to previous works in this group and the patterning method, the authors developed, is very simple but precise, applicable to various materials. The responsiveness of the involved photo-responsive supramolecular network and the tunability of the resulted reversible pattern are very nice. And these results are very interesting whatever from the perspective of the material chemistry, means to demonstrate diverse topological microstructures (such as macro-scale calligraphy pattern, QR pattern) or various potentials for smart displays, information storage, and anticounterfeiting of dynamic pattern indicated by authors.

In the end, I like the concise words and nice figures in the paper and I think it should be published. Hence, I recommend this manuscript for publications in Nature Communications.

Response: We thank you for the very positive and precious comments. The manuscript has been carefully revised based on the suggestions.

Comment 1: *I am curious why the CIE color space coordinates changed linearly demonstrated in Figure 3e. The authors mentioned it while seem to have no discussion on this interesting result.*

Response to 1: Thank you very much for your insightful consideration and constructive comment. We have added some discussion on the linear relationship of the CIE color space coordinates in the revised manuscript. Actually, the CIE color space coordinates are calculated from the fluorescence spectra. The fluorescence emission spectra of DSP-OH are shown in Fig. 2d and Fig. 4d. Each spectrum of DSP-OH with certain concentration of acid is the weighted sum of the spectra of pure DSP-OH and fully protonated DSP-OH, because it is the simple mixture of the pure DSP-OH and fully protonated DSP-OH, and there are no interaction between the components. As a result, the CIE color space coordinates exhibit similar characteristic with the fluorescence

spectra because it comes from the data of fluorescence spectra. Thus, each color of DSP-OH with certain concentration of acid is the weighted sum of the color of pure DSP-OH and fully protonated DSP-OH. Owing to the definition of the CIE color space, The point of the blend color must locate on the line of the individual two color points. Therefore, the CIE color space coordinates changed linearly demonstrated in Fig. 2e and Fig. 4e.

Comment 2: *Some scale bars of Figure 3f and Figure 4 look like absent, and I think the writing calligraphy and QR patterns should be macro-scale ones.*

Response to 2: Thanks for your professional suggestions. We have revised these figures with better resolution in the revised manuscript.

Comment 3: *In Figure 2e, the exposure time of 450 visible light under the photomask is unclear both in main text and annotation in figure.*

Response to 3: Thank you for your good advice. We have revised these figures and annotation with better resolution in the revised manuscript.

Comment 4: *There is a typo error “wrinting” in Figure 3f.*

Response to 4: Thank you for your careful check. We have modified the manuscript and the supporting information to correct the unsuitable terminologies and revised these figures with better resolution.

REVIEWERS' COMMENTS:

Reviewer #1 (Remarks to the Author):

This revised manuscript can be published as it is now.

Reviewer #2 (Remarks to the Author):

Using the advantages of supramolecular polymer networks, photochemistry and bilayer wrinkling systems, the authors demonstrated a dynamic dual pattern exhibiting the wrinkled topography and fluorescence in response to multiple stimuli, visible light and acid solution, that were not reported in previous studies.

However, in the application of anticounterfeiting insisted in this paper, it is required to generate the microstructure in which the wrinkle pattern can function as an identifier, and to extract and analyze the information according to the generated pattern.

It is difficult to say that this is a new anticounterfeiting technique by simply combining wrinkles and changing the fluorescence by chemical reaction.

In my opinion, it is very unlikely to be practically applied.

I do not recommend its publication.

Reviewer #3 (Remarks to the Author):

The authors have suitably addressed all the comments I raised in original review and the manuscript is acceptable for publication in Nature Communications.

Reviewer #1: This revised manuscript can be published as it is now.

Response: Thank you very much for your efforts and help related to this manuscript.

Reviewer #2: Using the advantages of supramolecular polymer networks, photochemistry and bilayer wrinkling systems, the authors demonstrated a dynamic dual pattern exhibiting the wrinkled topography and fluorescence in response to multiple stimuli, visible light and acid solution, that were not reported in previous studies.

Response: Thank you for the positive comments. In this work, we developed a simple and new strategy to realize dynamic dual pattern with responsive wrinkle and fluorescent, which might find potential application for anticounterfeiting.

Comment 1: *However, in the application of anticounterfeiting insisted in this paper, it is required to generate the microstructure in which the wrinkle pattern can function as an identifier, and to extract and analyze the information according to the generated pattern.*

Response to 1: Thank you for the comment. Actually, the wrinkled pattern exhibits potential application in multilevel anticounterfeiting, because it exhibits the characteristics of both traditional graphical tags and fingerprint-like structures and combines the advantages. On one hand, the graphical images formed by wrinkles, which could be identified by naked eyes due to the light scattering of wrinkle pattern, endows the capacity for serving as graphical encoded tag, just like what raised print, watermarks, fluorescent pattern and hologram do (*Nat. Rev. Chem.* **2017**, 1, 0031). For this tag, the graphical images serve as the identifiers, and we need to extract and analyze information hidden in the graphical images. On the other hand, the wrinkles exhibit fingerprint-like microstructures according to both the experiments and the liner bulking theory, and the microstructures are random and individual. These features enable the wrinkled pattern to serve as artificial fingerprint and the minutiae of wrinkled pattern such as ridge ending and bifurcation plays the role of the

identifier. In this case, the information is extracted from the fingerprint-like structures. Actually, the wrinkled pattern has the advantage over other graphical tags such as raised print and watermarks, due to the unique characteristic of fingerprint-like structures that the others do not have. It is the advantage but not the limitation that the encoded information in the fingerprint-like structures enhance the information capacity and security. However, we cannot ignore the importance of graphical images because of the fingerprint-like structures, and vice versa. Therefore, both the graphical images and fingerprint-like structures play an important role in the potential application for multilevel anticounterfeiting.

Comment 2: *It is difficult to say that this is a new anticounterfeiting technique by simply combining wrinkles and changing the fluorescence by chemical reaction.*

In my opinion, it is very unlikely to be practically applied.

Response to 2: Fluorescent patterns have been used in many fields for anticounterfeiting technologies in our daily life. Surface wrinkles exhibit potential application in anticounterfeiting due to its graphical pattern and biomimetic fingerprint-like structure (*Adv. Mater.* **2015**, 27, 2083-2089; *Nature* **2015**, 520, 164-165; *Nat. Rev. Chem.* **2017**, 1, 0031). However, either the fluorescent pattern or wrinkled tag faces a certain of risk of being cloned. To solve the problem, it is a feasible strategy to fabricate a new encoded tag by involving multiple anticounterfeiting technologies to single tag. Actually, in certain situation in our daily life, such as banknotes, tens of anticounterfeiting technologies are used together to enhance the information security and reduce the risk of being cloned (*Nat. Rev. Chem.* **2017**, 1, 0031). It is similar with our strategy but there is a difference. In the former case, each tag works individually and independently, and the risk of being cloned does not decrease for everyone. While, what we do is to fabricate a novel anticounterfeiting tag of multi pattern with the wrinkled topography and fluorescence color in response to multiple stimuli. Either the fluorescent pattern or the wrinkled topography is integrated into a single and individual tag, which will undoubtedly enhance the information capacity and security. Thus, it is a feasible strategy to fabricate a new

anticounterfeiting tag with dual pattern operated in multiple modes. However, it is still a challenge due to the complexity of the involved chemistry and material. In this study, we demonstrated a simple but precise patterning method, and the stimulus-responsive composite material is new compared to previous work. It is a progress in the dynamic surface due to its dual pattern and the multiple operating modes. This dual pattern shows potential application for anticounterfeiting. Although some limitations exist, we will do efforts to optimize the structure of the molecular and improve the system in the follow works, and the engineers and industries will help applying it to the products. In summary, it is a novel patterning method and the multi-responsive pattern with wrinkled topography and fluorescence exhibits potential application for anticounterfeiting.

Reviewer #3: The authors have suitably addressed all the comments I raised in original review and the manuscript is acceptable for publication in Nature Communications.

Response: Thank you very much for your efforts and help related to this manuscript.